# Recent Developments in CRISPR/Cas9 Genome-Editing Technology Related to Plant Disease Resistance and Abiotic Stress Tolerance

**DOI:** 10.3390/biology12071037

**Published:** 2023-07-22

**Authors:** İbrahim Erdoğan, Birsen Cevher-Keskin, Özlem Bilir, Yiguo Hong, Mahmut Tör

**Affiliations:** 1Department of Agricultural Biotechnology, Faculty of Agriculture, Kirsehir Ahi Evran University, Kırşehir 40100, Türkiye; ibrahim.erdogan@ahievran.edu.tr; 2Department of Biological Sciences, School of Science and the Environment, University of Worcester, Henwick Grove, Worcester WR2 6AJ, UK; ozlem.bilir@tarimorman.gov.tr (Ö.B.); y.hong@worc.ac.uk (Y.H.); 3Genetic Engineering and Biotechnology Institute, TÜBİTAK Marmara Research Center, Kocaeli 41470, Türkiye; birsen.keskin@tubitak.gov.tr; 4Trakya Agricultural Research Institute, Atatürk Bulvarı 167/A, Edirne 22100, Türkiye; 5Research Centre for Plant RNA Signaling, College of Life and Environmental Sciences, Hangzhou Normal University, Hangzhou 311121, China

**Keywords:** genome editing, CRISPR/Cas9, plant diseases, disease resistance, abiotic stress

## Abstract

**Simple Summary:**

Pests and diseases, along with environmental factors, significantly contribute to yield losses in crop production. Considering the detrimental impact of pesticides on both the economy and the environment, it is crucial to urgently develop methods that can prevent such damage. Additionally, it is imperative to address challenges posed by the growing world population, climate change, and the emergence of new pathogens. In this century, one of the most important advancements in terms of crop improvement lies in faster and more effective genome editing than is possible via traditional plant breeding, resulting in the production of transgene-free plant lines. The CRISPR/Cas9 genome-editing technique has emerged as the most widely used tool for creating plants with desirable traits, such as disease resistance and tolerance to abiotic stresses. These technologies enable the cultivation of crop plants capable of adapting to these new conditions, offering novel opportunities and solutions.

**Abstract:**

The revolutionary CRISPR/Cas9 genome-editing technology has emerged as a powerful tool for plant improvement, offering unprecedented precision and efficiency in making targeted gene modifications. This powerful and practical approach to genome editing offers tremendous opportunities for crop improvement, surpassing the capabilities of conventional breeding techniques. This article provides an overview of recent advancements and challenges associated with the application of CRISPR/Cas9 in plant improvement. The potential of CRISPR/Cas9 in terms of developing crops with enhanced resistance to biotic and abiotic stresses is highlighted, with examples of genes edited to confer disease resistance, drought tolerance, salt tolerance, and cold tolerance. Here, we also discuss the importance of off-target effects and the efforts made to mitigate them, including the use of shorter single-guide RNAs and dual Cas9 nickases. Furthermore, alternative delivery methods, such as protein- and RNA-based approaches, are explored, and they could potentially avoid the integration of foreign DNA into the plant genome, thus alleviating concerns related to genetically modified organisms (GMOs). We emphasize the significance of CRISPR/Cas9 in accelerating crop breeding processes, reducing editing time and costs, and enabling the introduction of desired traits at the nucleotide level. As the field of genome editing continues to evolve, it is anticipated that CRISPR/Cas9 will remain a prominent tool for crop improvement, disease resistance, and adaptation to challenging environmental conditions.

## 1. Introduction

Global food security faces multiple threats, including the challenges posed by a growing population, climate change, and the constant evolution and emergence of plant diseases [1,2,3]. By 2050, the world population is projected to exceed nine and a half billion, leading to a substantial increase in food consumption of 100–110%. However, current agricultural capacities suggest that crop yields for essential crops like wheat, corn, rice, and soybean will only witness a modest rise of 38–67% [4]. It is evident that global climatic changes, including escalating droughts, floods, and harmful micro-organisms, will adversely impact agricultural productivity [5,6]. Biologic stress on plants alone is anticipated to cause yield losses exceeding 40%, resulting in a 15% decline in the overall global food supply [7,8,9]. To address these challenges, pesticides, fertilizers, and other chemicals have been extensively used in recent years to enhance agricultural yield, promote plant health, and combat plant infections. However, the use of such chemicals poses significant threats to the environment, causing damage to soil, water, and vegetation. Additionally, they indirectly affect animals such as birds, fish, beneficial insects, and non-target plants [9,10,11,12,13,14].

Plant genomes have been modified using traditional plant breeding techniques or through physical (such as gamma radiation), chemical (such as ethyl methanesulfonate, EMS), and the biological (including T-DNA and transposon insertion) methods, resulting in point mutations, deletions, and gene duplications. While these approaches have led to crop improvement, they are time consuming, expensive, and often face challenges related to conventional breeding. Moreover, they can cause unintended rearrangements in the genome. Therefore, it is crucial to enhance the development of high-yielding crops that are disease-free and tolerant to abiotic stresses, enabling them to adapt to future challenges. Recognizing these challenges, the scientific community has long been committed to cultivating ideal crops. Genetically modified (GM) crops have been created by transferring beneficial genes to crops through a trans-kingdom approach. Consequently, scientists have directed their efforts toward developing genome-editing tools that can modify the genome without introducing transgenes [2,15,16,17].

The era of genome editing began with the introduction of zinc finger nucleases (ZFNs) and transcription activator-like effector nucleases (TALENs), though it reached new heights following the discovery of CRISPR/Cas technology. The development of CRISPR/Cas9 (Clustered Regularly Interspaced Short Palindromic Repeats/CRISPR-associated nuclease 9) has revolutionized genome editing in plants, enabling significant advancements beyond the outcomes that conventional breeding techniques can achieve [18]. The type II CRISPR/Cas9 system, which was initially found in *Streptococcus pyogenes*, has become the most widely adopted and extensively utilized system [19,20,21]. The importance of genome-engineering technologies in modern plant development and improvement cannot be overstated. The field of genome engineering has undergone a transformative revolution, which was largely driven by the development and widespread acceptance of CRISPR, which is recognized as one of the most potent gene-editing techniques available [22,23]. The CRISPR system functions by incorporating sequences from foreign elements into a small RNA-based memory, which serves as an inherited resistance mechanism. These short RNAs recognize the foreign invader and employ Cas proteins, which act as enzymatic units, to cut and eliminate the invader’s genetic material. In many aspects, the CRISPR system bears resemblance to the RNA-based defense mechanisms found in animal germ cells, which protect against mobile genetic material [24]. The CRISPR-Cas9 system has been simplified into two key components: a single chimeric RNA known as guide RNA (gRNA) and the enzyme Cas9. Together, these elements enable the editing of specific genomes and facilitate desired modifications in genome engineering [19]. The gRNA consists of CRISPR RNA (crRNA) and trans-activating CRISPR RNA (tracrRNA) sequences, which guide the sequence-specific cleavage of genomic DNA by Cas9 through a straightforward base-pairing process. Notably, gRNAs can be easily designed to recognize a target sequence of 20 nucleotides in length and induce precise Cas9-dependent cleavage of both DNA strands at a pre-defined location within the target, underscoring the utility of this technology.

The development of genome-edited plants using CRISPR/Cas9 technology consists of five steps: (i) selection of the target gene, (ii) designing a targeted gene-specific sgRNA, (iii) assembling Cas9 and sgRNA, (iv) transformation to the target plant, and (v) regeneration and screening of plant lines (see Figure 1). Here, we describe the usage and benefits of CRISPR genome-editing technology for developing plant lines that are resistant to disease and abiotic stress and provide examples of genes edited to enable crop improvement.

## 2. Genome Editing Tools and Their Comparison

Genome editing techniques rely on three main sequence-specific nuclease systems: zinc-finger nucleases (ZFNs), transcription activator-like effector nucleases (TALENs), and the Clustered Regularly Interspaced Short Palindromic Repeats/CRISPR-associated protein (CRISPR/Cas). Mega-nucleases, such as ZFNs and TALENs, have recently enabled the targeted modification of specific genome sequences [25,26]. Until 2013, ZFNs were the most commonly used genome editing technique, followed by TALENs [27,28]. These methods involve engineered fusion proteins, in which a DNA binding domain is fused to the non-specific nuclease domain of the restriction enzyme FokI. They have been successfully applied in various species, including plants [29,30]. However, ZFNs and TALENs face challenges as they require the introduction of a new protein after target verification, making their application more complex [9]. Consequently, these methods have not been widely adopted in plant genome editing, prompting scientists to explore alternative approaches [31,32]. Unlike ZFNs and TALENs, which require specific proteins and sequences, the CRISPR/Cas system only requires a guide RNA (gRNA), which offers a significant advantage (refer to Table 1). The CRISPR/Cas system has emerged as a superior gene-editing technique due to its simplified requirements. In the ZFN and TALEN systems, the restriction endonuclease FokI contains a catalytic domain that generates double-strand breaks with sticky ends of varying lengths, which depend on the linker and spacer used. On the other hand, the Cas9 system of CRISPR/Cas comprises two cut site—RuvC and HNH—that generate blunt ends by cleaving the target DNA three nucleotides upstream of the protospacer adjacent motif (PAM) [19]. The CRISPR target site requires a 3-base pair protospacer adjacent motif (PAM) situated at the 3′ NGG end of the 20-nucleotide recognition sequence. When the Cas9 enzyme generates double-strand breaks (DSBs), initiating the DNA repair mechanism, two biological mechanisms can be employed for genome editing. The first mechanism—non-homologous end joining (NHEJ)—is an error-prone process that leads to small insertions and deletions, compromising the functionality of the cut sites. The second approach is homology-directed repair (HDR), which utilizes templates created via homologous DNA sequences for repair purposes. This repair mechanism can be harnessed to precisely modify the genome or introduce foreign DNA using an externally constructed template donor [33]. The CRISPR/Cas9 method overcomes challenges associated with complex construction of DNA-binding domain expression cassettes, reduced target sensitivity, and lower cutting efficiency encountered in ZFN and TALEN gene-editing technologies. Furthermore, the CRISPR/Cas9 system is easier and more practical to implement [34,35]. In contrast, gRNA-guided cutting in the CRISPR/Cas system relies on a simple base-pairing mechanism, eliminating the need for complex and labor-intensive protein engineering for each target. Instead, a 20-nucleotide gRNA sequence can be designed to specifically recognize and bind to the desired target DNA sequence [3]. However, it should be noted that the CRISPR/Cas system’s requirement for a protospacer adjacent motif (PAM) at the target recognition site may impose limitations on its applicability in certain gene-editing scenarios [36]. In light of the potential drawbacks of a PAM-free nuclease, there are several avenues along which this field can proceed. One approach is to focus on other Cas nucleases, such as Cas9, Cas12a, and remaining Cas nucleases, which still have room to relax PAM recognition without completely eliminating it. By combining ortholog mining and PAM engineering, researchers can expedite the development of these nucleases. The abundance of characterized Cas9 nucleases suggests that there is a wide diversity of PAM sequences in nature that has yet to be fully explored. For example, Type V-C nucleases recognize PAMs with as little as a single base, making them promising candidates for PAM relaxation. Determining the structures of nucleases that naturally recognize only one nucleotide could reveal new insights into PAM recognition and guide future engineering efforts [37]. Type I CRISPR-Cas10 is a CRISPR system that has been shown to induce small insertions and deletions (indels), as well as bi-directional long-range deletions, in tomatoes (*Solanum lycopersicum* L.). This system allows precise modifications in the tomato genome, with deletions spanning up to 7.2 kb in length. This capability creates the potential for targeted genetic modifications and the development of improved tomato varieties. On the other hand, Type IV CRISPR-Cas13 is a distinct CRISPR system that targets RNA instead of double-stranded DNA. This unique feature enables highly specific knockdown of target genes at the RNA level. By utilizing the RNA-targeting capability of Cas13, researchers can achieve precise regulation of gene expression and potentially develop novel approaches for various applications.

Both Type I CRISPR-Cas10 and Type IV CRISPR-Cas13 demonstrate the versatility and potential of CRISPR technologies for precise genome editing and gene regulation in different organisms, including important crops, such as tomatoes. These advancements open up new possibilities for crop improvement and functional genomics [38,39,40]. In the field of genome engineering in plants, CRISPR activation systems have significantly advanced the capabilities of targeted mutagenesis, base editing, and gene activation. However, these tools are typically used independently, limiting their potential for combined applications. To address this limitation, researchers developed a versatile platform called CRISPR-Combo, which utilizes a single Cas9 protein to enable simultaneous genome editing and gene activation in plants [41]. The CRISPR-Combo platform offers powerful applications to enhance plant genome editing. Firstly, they demonstrate its utility in shortening the plant life cycle and simplifying the screening process for transgene-free genome-edited plants. This outcome is achieved by activating a florigen gene in *Arabidopsis*, leading to accelerated flowering and seed production. By activating morphogenic genes in poplar, they achieve accelerated plant regeneration and propagation, reducing the time and effort required to generate a large number of edited plants. This novel approach allows the efficient enrichment of heritable targeted mutations, providing a valuable tool for crop breeding. In summary, CRISPR-Combo represents a versatile genome engineering tool that has promising applications in crop breeding [41].

## 3. Arise of CRISPR/Cas9 Technology

In 1987, the CRISPR/Cas system was discovered in prokaryotes and found to be an adaptive immune system that defends against invading bacteriophages or plasmids. Researchers studying *Escherichia coli* identified approximately 32 nucleotide non-repeat sequences and tandem repeats downstream of the *iap* gene. These tandem repeats were later named CRISPR in the year 2000 [42,43]. Studies revealed that the CRISPR spacer sequences showed similarity to sequences from exogenous sources, such as bacterial plasmids and phages. When a bacteriophage infects a bacterium, the endogenous CRISPR system replicates the repeat regions of the phage genome, which are then separated from the protospacer adjacent motif (PAM) through spacer sequences [44,45,46]. The bacterium’s CRISPR/Cas system detects the viral genome and neutralizes it during subsequent phage attacks, functioning as the bacterium’s immune system. The Cas library in the bacterium preserves a record of the invading viral sequences and aids in their destruction upon re-attack.

In traditional crop improvement methods, mutation breeding methods are often employed using chemicals like ethyl methanesulfonate (EMS) or gamma radiation [47]. However, the CRISPR/Cas9 system offers the ability to modify single or multiple target genes. Cas9, which is the enzyme from the type II CRISPR-Cas system of *S. pyogenes*, is a large monomeric DNA nuclease. It is guided by two non-coding RNA complexes—CRISPR RNA (crRNA) and trans-activating CRISPR RNA (tracrRNA)—that cleave the DNA target region next to the PAM sequence motif [19,48,49]. Cas9 possesses two nuclease domains that are similar to the RuvC and HNH nucleases of the Cas9 protein (Figure 2).

Indeed, the Cas9 protein, due to its HNH and RuvC-like nuclease domains, is responsible for cleaving the DNA target site. The HNH domain cleaves the complementary DNA strand, while the RuvC-like domain cleaves the non-complementary strand, resulting in a blunt cut in the target DNA [19]. When the Cas9 protein is combined with a single guide RNA (sgRNA) and targeted to a specific genomic site, it induces site-specific double-strand breaks (DSBs) in the DNA of living cells across various organisms. Following the generation of DSBs, different intracellular repair mechanisms come into play, leading to various genome modifications. The two primary repair pathways are non-homologous end-joining (NHEJ) and homology-directed repair (HDR) in nature. NHEJ is an error-prone repair mechanism that often results in random insertions and deletions (indels) at the target gene site. These indels can disrupt the functioning of the gene, leading to gene knockout or loss-of-function mutations. On the other hand, HDR relies on the use of a template, which is typically a homologous DNA sequence, to repair the DSB with high precision. This repair pathway can be utilized to introduce desired modifications or specific DNA sequences into the genome. It is important to note that the repair outcome depends on the specific repair mechanisms of the organism and the factors influencing repair pathway choice. NHEJ is the predominant pathway in many organisms, while HDR is typically less efficient but can be enhanced by optimizing the experimental conditions.

Base editing and prime editing are two recent advancements in CRISPR/Cas technology that have revolutionized the field of genome editing. These techniques offer the ability to induce precise point mutations in the DNA without the need to induce double-strand breaks (DSBs), which can lead to unintended mutations. Base editing encompasses two main types of editor: cytosine base editors (CBEs) and adenine base editors (ABEs). CBEs utilize a modified Cas9 protein fused with an enzyme that is capable of chemically modifying cytosine to induce specific nucleotide changes. This process allows targeted conversion of cytosine into thymine (C-to-T) or guanine (C-to-G) mutations, depending on the desired outcome. On the other hand, ABEs utilize a modified Cas9 protein fused with an enzyme that can convert adenine to inosine, which is then recognized as guanine during DNA replication. This process enables the induction of adenine to guanine (A-to-G) mutations. Prime editing is a more advanced genome editing technique that expands the scope of modifications that can be achieved. It combines a modified Cas9 protein with a reverse transcriptase enzyme and a prime-editing guide RNA (pegRNA). The pegRNA contains the desired edits in the form of a template, which is reverse transcribed and integrated into the target DNA site, allowing precise changes in the DNA sequence. Prime editing enables a broader range of modifications, including transitions, transversions, insertions, and deletions, providing greater flexibility in genome engineering. These advancements in base editing and prime editing have greatly expanded the toolbox of CRISPR/Cas technology, allowing more precise and versatile genome modifications. They offer exciting possibilities for targeted genetic changes and the development of improved crops [50]. Recent studies have highlighted the significance of utilizing endogenous RNA Pol III promoters, specifically U3 and U6 promoters, in the CRISPR/Cas9 system to enhance genome editing efficiency in various plant systems. These promoters are responsible for transcribing single or multiple guide RNAs that guide the Cas9 nuclease to target specific genomic regions. By utilizing endogenous RNA Pol III promoters, researchers have observed improved editing efficiency and precision in plant genome editing. The use of species-specific U3/U6 promoters holds promise in terms of advancing the field of genome editing by enabling more specific and efficient targeting of desired genomic sequences [51].

## 4. CRISPR/Cas-Mediated Genome Editing for Disease Resistance

Plant diseases caused by biotic factors, such as viruses, fungi, oomycetes, and bacteria, pose a significant threat to crop productivity and global food security. These infections result in substantial yield losses and quality reduction in field crops, fruits, and other edible plant materials. Approximately 20 to 40% of worldwide crop losses are attributed to biotic factors [52]. Traditionally, plant disease resistance has been achieved through the introgression of resistance genes (*R*-genes) from wild relatives of the cultivated crop. The most common type of resistance mechanism involves nucleotide-binding leucine-rich repeat (NB-LRR) proteins, which recognize specific products of pathogen *Avirulence* (*Avr*) genes [53]. Upon recognition, a defensive response is triggered, leading to the host cell’s programmed cell death; this process is known as the hypersensitive reaction (HR) [53]. Another approach to disease resistance is the suppression of susceptibility (*S*) genes, which are essential for pathogen infection [54]. Many pathogens rely on specific host genes for successful infection and proliferation. Biotrophic fungal pathogens, such as powdery mildews, require prolonged interaction with host cells to achieve effective proliferation [55]. The expression of specific host genes, known as *S*-genes or susceptibility genes, is necessary for pathogen recognition, penetration, evasion of host defenses, and fulfillment of the pathogen’s metabolic and structural needs [55]. Mutations in *S*-genes can result in long-term and broad-spectrum recessive heritable resistance [56,57]. Therefore, *S*-genes play a critical role in plant–pathogen interactions, influencing the host’s susceptibility to infection. The functions of *S*-genes can be categorized into three major molecular mechanisms: basic compatibility, sustained compatibility, and negative modulation of plant immune signals. Basic compatibility involves host recognition and pathogen penetration, sustained compatibility is necessary for pathogen proliferation and dissemination, and negative modulation of plant immune signals helps to suppress host defense responses [55]. Understanding the molecular mechanisms that underlie *S*-gene functions and their interactions with pathogens is crucial to the development of effective strategies that enhance plant disease resistance. The barley *Mlo* (*mildew resistance locus*) gene is a well-known example of an *S*-gene mutation that confers resistance to powdery mildew in barley. Deployment of loss-of-function *Mlo* alleles in barley has resulted in the development of powdery mildew-resistant barley varieties. These resistant plants have been cultivated in the field for many years, demonstrating the durability of *S*-gene-based resistance against virulent powdery mildew strains [58]. The cloning of the barley *Mlo* gene revealed that it is conserved across the plant kingdom and exists as a multi-copy gene family in higher plants [59,60,61]. Studies have shown that mutation of the *Mlo* gene confers resistance to powdery mildew. The *Mlo* protein is essential for the penetration of powdery mildew fungal spores into host epidermal cells. The *Mlo* gene encodes a membrane-associated protein with seven transmembrane domains, and its mutation, which uses the CRISPR-Cas9 system, has resulted in the generation of transgene-free and powdery mildew-resistant plants [62,63,64]. *Mlo*-like genes have also been implicated in powdery mildew susceptibility in various plant species, including *Arabidopsis*, tomato, pea, pepper, tobacco, bread wheat, and potentially grapevine and peach [63,65,66,67,68,69,70]. These genes are categorized into different classes based on their phylogenetic relationships. Class IV contains *Mlo*-like genes associated with powdery mildew susceptibility in monocotyledonous species, while class V contains those found in dicotyledonous species. Other classes (I, II, III, and VI) include *Mlo*-like genes that have not yet been identified as *S*-genes [61]. The development of next-generation sequencing and CRISPR technologies has facilitated the identification and targeting of *S*-genes involved in the enhancement of disease resistance in plants. CRISPR/Cas9 has been used to target *S*-genes, including the *Mlo* gene, to confer protection against important plant pathogens. Modifying these *S*-genes can disrupt the compatibility between the host and the pathogen, leading to broad-spectrum and durable disease resistance [71]. Another example is the targeting and alteration of the Enhanced Disease Resistance 1 (*EDR1*) gene, resulting in a significant decrease in powdery mildew in wheat [35]. One of the notable advantages of CRISPR/Cas9 technology is its ability to target multiple genes simultaneously using a single construct. Studies have shown that a single molecular structure in *Arabidopsis* can simultaneously induce mutations in 14 different genes [72]. Multiplex genome-editing techniques, through which multiple guide RNAs (gRNAs) are integrated into a single construct under the control of a U3 or U6 promoter, have been successfully applied in crops [73,74,75,76,77]. Previous methods for transgene removal involved either molecular excision or segregation via selfing or backcrossing to the original parent line. The Cas9 system allows transgene-free lines to be selected from segregating populations once a successful mutation in the target genomic region is confirmed [71,78,79].

Viruses pose a significant biotic stress risk to plants and can cause diseases in various commercially important crops. The Eukaryotic Translation Initiation Factor 4E (eIF4E) is a key protein required for the infection cycle of Potyviridae family viruses, which are single-stranded positive-sense RNA viruses. The viral protein genome-linked (VPg) element at the 5′-terminal of potyviruses interacts with eIF4E, and disrupting this interaction has been shown to confer immunity against potyviruses in different plant species [80]. Using CRISPR-Cas9 technology, researchers have successfully disrupted *eIF4E* genes to generate resistance against potyviruses and ipomoviruses in *Arabidopsis* and *Cucumis sativus* (cucumber) in independent studies [18,78]. Importantly, these genome-edited plants did not contain transgenes, demonstrating the potential of CRISPR technology to produce virus-resistant crops without the need to introduce foreign DNA [18,78].

In addition to viral infections, CRISPR-Cas systems are being investigated regarding their potential role in combating bacterial infections in agriculture. One example is citrus canker, which is a devastating disease caused by *Xanthomonas citri* subsp. *citri*, which leads to significant yield losses in citrus production worldwide [81]. The CRISPR/Cas9 method has been employed to modify the *OsSWEET13* gene, resulting in resistance to bacterial blight caused in rice by the γ-proteobacterium *Xanthomonas oryzae* pv. *oryzae* [82]. *OsSWEET13* is an *S*-gene that encodes a sucrose transporter involved in plant–pathogen interactions [73]. Table 2 provides examples of CRISPR-mediated editing of *S*-genes to generate disease-resistant crops. These studies highlight the potential of CRISPR technologies in addressing various plant diseases caused by viral and bacterial infections.

## 5. CRISPR/Cas-Mediated Genome Editing for Abiotic Stress Tolerance

Abiotic stresses, such as drought, heat, cold, and salt stress, pose significant challenges to crop production worldwide. Plants have evolved complex mechanisms that allow them to tolerate and respond to these stresses, which involve cellular and transcriptional regulation. Severe stress conditions can lead to membrane damage, cellular injury, and visible symptoms of necrosis in plants [109]. The advent of genome-editing technologies has provided opportunities for researchers to explore the tolerance mechanisms and develop novel traits for abiotic stress resistance. Abiotic stress tolerance is a complex trait controlled by multiple genes, making it challenging to manipulate through traditional breeding methods alone [110]. Numerous regulatory and structural genes are involved in abiotic stress responses. By focusing on “*Tolerance* (*T*) genes” that exhibit positive regulatory responses to stress, researchers can enhance stress tolerance through approaches such as gene overexpression, modification of promoter elements, modification of upstream Open Reading Frames (uORFs), or CRISPR activation [111]. Conversely, “*Sensitive* genes (*S*)” that are involved in stress susceptibility can also be targeted to study abiotic stress resistance. CRISPR-mediated knockout studies of *S*-genes have generated plants with enhanced resistance to abiotic stresses [112]. By disrupting these genes, researchers can uncover their functions in stress sensitivity and potentially develop crops with improved stress tolerance. The use of CRISPR technology enables precise and targeted modifications of specific genes involved in abiotic stress responses. By manipulating *T*-genes or disrupting *S*-genes, researchers can gain insights into the underlying mechanisms of stress tolerance and develop crop varieties that have enhanced resilience to abiotic stresses.

### 5.1. Drought Tolerance

Water scarcity and drought stress have become significant global challenges that affect both developing and developed countries. With the increasing impact of global warming, water evaporation from the Earth has intensified, leading to drought stress in plants. Drought stress has adverse effects on plant morphology and biochemistry, resulting in significant crop losses [113,114]. The CRISPR/Cas gene-editing technology has been employed to enhance drought tolerance in various plant species. By targeting specific genes involved in drought-response pathways, researchers have successfully improved the drought tolerance of crops. Here are some examples: *Dehydration responsive element binding 2* (*TaDREB2*) and *Ethylene responsive factor 3* (*TaERF3*) were previously edited using CRISPR/Cas in wheat, resulting in enhanced drought tolerance [115]. A loss-of-function mutation of the *SAPK2* gene through CRISPR/Cas editing has improved drought tolerance in *Oryza sativa* (rice) by affecting ABA signaling, where *SAPK2* acts as a primary mediator [116]. CRISPR/Cas9-mediated editing of the *SLNPR1* gene in tomato led to enhanced drought tolerance, as evidenced by improved leaf retention under drought stress [117]. The *OsDST* gene edited using CRISPR/Cas in an *O. sativa* cultivar MTU1010 improved drought and salt tolerance by promoting leaf retention under drought stress conditions [118,119]. Editing of the *OsERF109* gene using the CRISPR/Cas system increased abiotic stress tolerance in *O. sativa* cultivars [120]. Several *ERF* family members, including *OsBIERF1*, *OsBIERF3*, and *OsBIERF4*, were edited using CRISPR/Cas in *O. sativa*, resulting in enhanced abiotic stress tolerance [97]. In maize, CRISPR/Cas9-mediated editing of the ethylene response factor *ARGOS8* improved drought tolerance [121]. Knock-out of the *ZmWRKY40* gene in *Zea mays* (maize) using CRISPR/Cas technology led to increased tolerance of drought stress [122]. In tomato, CRISPR/Cas knock-out of the *Mitogen-Activated Protein Kinase 3* (*SlMAPK3*) gene resulted in drought tolerance, which was characterized by increased levels of malondialdehyde, proline, and H_2_O_2_ in the mutant lines [123]. Knock-out of the *SlNPR1* gene using CRISPR/Cas9 in tomato reduced drought resistance and down-regulated drought-related genes [117]. These examples highlight the potential of CRISPR/Cas gene editing in improving drought tolerance in various crop species by targeting the specific genes involved in drought-response pathways. By modifying these genes, researchers aimed to enhance the ability of plants to withstand drought stress and minimize crop losses.

### 5.2. Salt Tolerance

Salt stress can have detrimental effects on plant cells, ranging from ion disturbances to necrosis [124]. It triggers various cellular changes, including the production of secondary signal molecules (such as ROS), synthesis of abscisic acid (ABA), alterations in calcium levels and calcium/calmodulin-dependent kinase activation, and the activation of salt overly sensitive (SOS) homeostatic signaling pathways [124]. The CRISPR/Cas gene editing system has been employed to enhance salt stress resistance in several genes. Here are some examples: *OsBBS1* (*Bilateral Blade Senescence 1*) and *OsMIR528* (*microRNA528*) were identified as positive regulators of salt stress and early leaf senescence in *O. sativa* (rice), respectively. CRISPR/Cas-based targeted mutations in these genes improved salt tolerance in rice [118,125]. Inducing the expression level of the *OsRAV2* (related to *ABI3/VP1 2* gene) gene through CRISPR/Cas-based targeted mutations enhanced salt tolerance in *O. sativa* [126]. Loss-of-function mutations of the *SnRK2* (*SNF1-related protein kinase 2*) and *SAPK-1* and SAPK-*2* (*Osmotic Stress/ABA-Activated Protein Kinases*) genes using CRISPR in rice resulted in increased salinity resistance [126]. Knockout of the *SlMAPK3* (*Mitogen-Activated Protein Kinases 3*) gene in tomato led to decreased expression levels of *SlLOX* (*Lipoxygenases*), *SlGST* (*Glutathione S-transferase*), and *SlDREB* (*Dehydration-Responsive Element-Binding* ) genes, resulting in improved salt tolerance [123]. Overexpression of *GmMYB118* (transcription factors) using CRISPR approaches enhanced drought and salt tolerance in soybeans and *Arabidopsis* [127]. Editing of the *SAPK1* and *SAPK2* genes in *O. sativa* increased salt stress tolerance [128]. Knockout of the *SlARF4* (*Auxin Response Factors 4*) gene, which negatively regulates salt tolerance and osmotic stress, improved salt tolerance in tomato [129]. In *Arabidopsis*, knockout of the *AtC/VIF1* (*Cell Wall/Vacuolar Inhibitor*) gene, which affects ABA response, conferred salt tolerance [130]. In *O. sativa*, knockout of the *OsDST* (*Drought and Salt Tolerance*) gene, which affects stomata density and leaf thickness, resulted in increased drought and salt tolerance [119]. These examples demonstrate the application of CRISPR/Cas gene editing to the improvement of salt stress tolerance in various plant species by targeting specific genes involved in salt stress response pathways. By manipulating these genes, researchers aim to enhance plants’ ability to cope with salt stress and improve their overall salt tolerance.

### 5.3. Cold Tolerance

The *C-repeat Binding Factor 1* (*CBF1*) plays a crucial role in protecting plants from cold/chilling injury and preventing electrolyte leakage [131]. Mutant tomato lines with altered *CBF1* expression exhibited increased accumulation of hydrogen peroxide and indole acetic acid, which contributed to enhanced cold tolerance [131]. Annexin (*OsANN3*), which is a gene that encodes a calcium-dependent phospholipid binding protein, has been identified as a contributor to cold stress tolerance in rice. CRISPR/Cas9 editing of the *OsANN3* gene resulted in increased relative electrical conductivity and improved cold tolerance [132]. The *Stress/ABA-Activated Protein Kinase 2* (*SAPK2*) gene-edited rice, which was generated using CRISPR/Cas9, showed resistance to cold stress [116]. In *O. sativa* (rice), the OsMYB30 mutant created through CRISPR/Cas gene editing exhibited enhanced cold tolerance, with an efficiency of approximately 63% [133]. These studies demonstrate the potential of CRISPR/Cas gene editing in improving cold stress tolerance in plants by targeting genes such as *CBF1*, *OsANN3*, *SAPK2*, and *OsMYB30*. Manipulating these genes can enhance plants’ ability to withstand low temperature conditions and minimize the damage caused by cold stress.

### 5.4. Heat Tolerance

Heat stress triggers various responses in plants, including alterations to heat shock proteins, enzymes involved in reactive oxygen species (ROS) synthesis, and genes that encode scavenger proteins [134]. Through CRISPR/Cas gene editing, several heat stress-related genes have been targeted to understand their roles in heat tolerance and improve thermotolerance in plants. Deletion mutants of *Heat Stress-Sensitive Albino 1* (*HSA1*) generated using CRISPR/Cas exhibited increased sensitivity to heat stress compared to wild-type tomato plants, indicating the involvement of *HSA1* in heat tolerance [135]. In tomato, thermotolerance was achieved by editing *BZR1* (*Brassinazole-Resistant 1*) expression using CRISPR technology. Mutant *BZR1* tomato lines displayed impaired hydrogen peroxide (H_2_O_2_) production in the apoplast and a reduction in the stimulation of *Respiratory Burst Oxidase Homolog 1* (*RBOH1*), indicating the importance of *BZR1* in influencing heat stress response [136]. CRISPR/Cas-mediated editing of the *Agamous-Like 6* (*AGL6)* gene in tomato resulted in the development of heat-tolerant plants that exhibited parthenocarpic fruit formation [137]. In maize, the mutation of the *Thermosensitive Genic Male Sterile 5* (*TGMS5*) gene using CRISPR/Cas technology allowed the production of thermosensitive male sterile plants [138]. These examples highlight the potential application of CRISPR/Cas gene editing to help us understand the function of heat stress-related genes and develop heat-tolerant traits in plants. By manipulating these genes, researchers aim to enhance the ability of plants to withstand high-temperature conditions and minimize the negative effects of heat stress on crop productivity. Table 3 provides further information about abiotic stress-related genes editing performed using CRISPR/Cas technology.

## 6. Conclusions

Mutation breeding based on physical radiation and chemically induced random mutagenesis has been widely used in the past for genome engineering. However, the emergence of CRISPR/Cas9 genome-editing technology has opened new possibilities and revitalized this field. With the integration of next-generation sequencing technologies, CRISPR/Cas9 has the potential to contribute to the development of future crops and mitigate the negative impacts of climate change on global food production. One of the primary concerns of genome-editing research is off-target effects. Despite being designed for specific genomic regions, genome modification tools can sometimes bind to unintended locations, resulting in undesired alterations. Off-target mutations are typically caused by the presence of other sequences in the genome that are identical or similar to the target gene’s sequence. Various computational methods have been developed to predict potential off-target sites in the genome [180,181]. While some studies have reported off-target effects and questioned the specificity of the CRISPR/Cas9 system, other studies have shown no off-target effects [182,183]. To address off-target activity, different strategies have been employed. Examples include using shorter guide RNAs (sgRNAs) of less than 20 nucleotides and dual Cas9 nickases, which create single-strand breaks, reducing the likelihood of off-target effects [184,185,186]. One of the most significant advantages of the CRISPR/Cas9 system is its ability to simultaneously target homologous genes with a single sgRNA [180,186]. Additionally, the Cas9/sgRNA expression vectors can contain multiple sgRNAs, enabling the study of gene families and their pathways [75,180]. Efforts have been made to deliver Cas9 and sgRNA into cells as protein and RNA to prevent the integration of foreign DNA into the genome. Studies have successfully delivered Cas9 protein and sgRNA directly to plant protoplasts using polyethylene glycol (PEG), as well as biolistically delivered the Cas9-sgRNA ribonucleoprotein complex to maize embryos, resulting in targeted mutations in regenerated plants [187,188]. By avoiding the use of recombinant DNA that can integrate into the genome, plants generated through these methods can be exempt from GMO regulations. These genome-editing techniques offer the potential to introduce desired modifications at the nucleotide level, such as improving yield and quality and providing resistance to diseases, pests, and abiotic stresses without the need for gene transfer. They also have the advantage of saving time and resources compared to labor-intensive processes like traditional selection and backcrossing. In the field of genome editing, there has been a shift from early approaches that relied on mutagenic repair of induced double-strand breaks to more precise and pre-defined modifications. This progress has been achieved through constant optimization of the tools used in genome editing. One area of advancement is the development of base editors, which are more efficient in inducing specific base changes in the genome. These base editors have been improved to have enlarged editing windows, allowing a broader range of targeted modifications. Additionally, advancements in base-editing techniques have enabled the previously challenging C-G transversions, expanding the repertoire of possible genetic modifications. Another significant development is the introduction of prime editors, which have been optimized for applications in plants. Prime editors offer greater precision by allowing the induction of specific substitutions, insertions, and deletions into the genome. This precise control over genetic modifications enhances the potential for targeted improvements in crops. Furthermore, recent breakthroughs have focused on precise restructuring of chromosomes. This approach enables not only the breakage or formation of genetic linkages, but also the swapping of promoters. By manipulating chromosomal structures, researchers can achieve more intricate modifications in the genome, opening up new possibilities for gene regulation and functional genomics. Overall, the ongoing optimization and refinement of genome editing tools have led to significant progress in achieving precise and pre-defined modifications in plant genomes. These advancements hold great promise for crop improvement, enabling the development of plants with desired traits, enhanced productivity, and improved resistance to biotic and abiotic stresses [189]. Considering the relatively short history of next-generation genome-editing techniques, it is evident that they have a high developmental capacity and significant innovation potential in the fields of breeding and plant health. Plants developed using CRISPR/Cas9 genome-editing technology may be more readily accepted by society, as they do not contain foreign genetic material transferred from other organisms. These genome-editing technologies will continue to be used as they are valuable tools in terms of crop improvement, disease resistance, and tolerance of abiotic conditions due to their ability to generate desired mutations in targeted gene regions.

## Figures and Tables

**Figure 1 biology-12-01037-f001:**
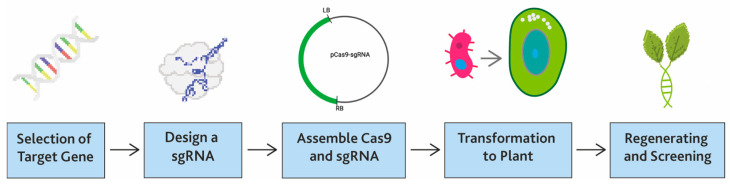
The basic flow of CRISPR/Cas9 technology used to edit target genes.

**Figure 2 biology-12-01037-f002:**
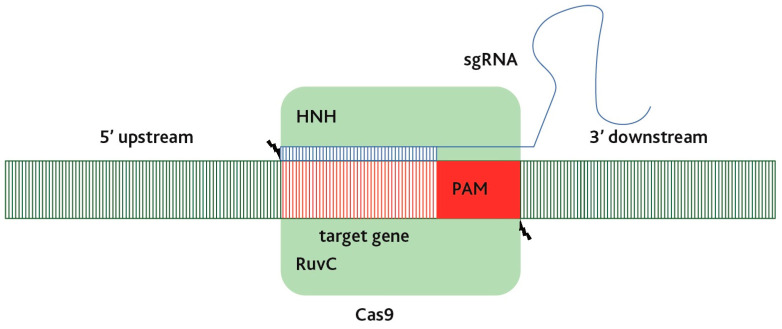
RNA-guided cleavage by the Cas9 protein.

**Table 1 biology-12-01037-t001:** Specifications of genome editing (GE) tools.

Genome Editing Tools	Target Site (bp)	Off Targeting	Enzyme	DNA Binding Mediator	Binding Specifity	DNA Cleavage	Usage	Origin
**CRISPR/Cas9**	20	Variable	Cas9	crRNA/ sgRNA	1:1 nucleotide pairing	RNA-dependent	Easy	Bacteria/Archaea
**ZFNs**	18–36	High	FokI	Zinc-finger protein	3 nucleotides	Protein-dependent	Highly difficult	Eukaryotes
**TALENs**	30–40	Low	FokI	Transcription activator-like effector	1 nucleotide	Protein-dependent	Difficult	Bacteria

**Table 2 biology-12-01037-t002:** Genes targeted by CRISPR-based genome editing techniques for imparting resistance against diseases.

Host Plant	Pathogen	Disease	Targeted Gene	Delivery Method	Transgene-Free	Result	References
**Apple**	*Erwinia amylovora*	Fire blight	*DIPM-1*, *DIPM-2* and *DIPM-4*	*Agrobacterium*-mediated transformation	Yes	Enhanced disease resistance	[83]
**Arabidopsis**	*Oidium neolycopersici*	Powdery mildew	*PMR4*	*Agrobacterium*-mediated transformation	No	Enhanced disease resistance	[84]
	Beet Severe Curly Top Virus (BSCTV)	DNA viral disease	IR, CP, Rep	*Agrobacterium*-mediated transformation	No	Geminivirus-resistant plants	[85]
	Turnip Mosaic Virus (TuMV)	RNA viral disease	*Elf(iso)4E*	*Agrobacterium*-mediated transformation	Yes	Potyvirus-resistant plants	[78]
**Banana**	Banana Streak Virus (BSV)	DNA viral disease	eBSV	*Agrobacterium*-mediated transformation	Not defined	Inactivation of eBSV caused asymptomatic plants	[86]
**Barley**	Wheat Dwarf Virus (WDV)	DNA viral disease	MP, CP, Rep/Rep, IR	*Agrobacterium*-mediated transformation	No	No disease symptoms and virus presence	[87]
**Cacao**	*Phytophthora tropicalis*	Black pod rot	*TcNPR3*	*Agrobacterium*-mediated transformation	No	Enhanced disease resistance	[88]
**Citrus**	*Xanthomonas citri* subsp. *citri*	Citrus canker	*CsLOB1*	*Agrobacterium*-mediated transformation	No	Enhanced disease resistance	[89]
	*X. citri* subsp. *citri*	Citrus canker	*CsLOB1*/promoter	*Agrobacterium*-mediated transformation	No	Enhanced disease resistance	[90]
**Cucumber**	Cucumber Vein Yellowing Virus (CVYV), Zucchini Yellow Mosaic Virus (ZYMV), and Papaya Ring Spot Mosaic Virus-W (PRSV-W)	RNA viral disease	*elf4E*	*Agrobacterium*-mediated transformation	Yes	Resistance to viruses	[18]
**Grapevine**	*Erysiphe necator*	Powdery mildew	*Mlo-7*	Polyethylene glycol-mediated (PEG) protoplast transformation	Yes	Enhanced disease resistance	[83]
	*Botrytis cinerea*	Gray mold	*VvWRKY52*	*Agrobacterium*-mediated transformation	No	Enhanced disease resistance	[91]
**Papaya**	*P. palmivora*	Root, stem, and fruit rot	*alEPIC8*	*Agrobacterium*-mediated transformation	Not defined	Enhanced disease resistance	[92]
**Rice**	*X. oryzae* pv. *Oryzae*	Bacterial Blight	*SWEET11*, *SWEET13* and *SWEET14*/promoter	*Agrobacterium*-mediated transformation	Not defined	Enhanced broad-spectrum disease resistance	[93]
	*X. oryzae* pv. *Oryzae*	Bacterial Blight	*Os8N3*/promoter	*Agrobacterium*-mediated transformation	Yes	Enhanced disease resistance	[94]
	*X. oryzae* pv. *Oryzae*	Bacterial Blight	*OsSWEET11* and *OsSWEET14*/promoter	*Agrobacterium*-mediated transformation	No	Enhanced broad-spectrum disease resistance	[95]
	*X. oryzae* pv. *Oryzae*	Bacterial Blight and Rice Blast	*Xa13*	Not defined	Yes	Enhanced disease resistance	[96]
	*X. oryzae* pv. *Oryzae*	Bacterial Blight	*OsSWEET13*	*Agrobacterium*-mediated transformation	No	Enhanced disease resistance	[82]
	*Magnaporthe oryzae*	Rice blast	*OsERF922*	*Agrobacterium*-mediated transformation	Yes	Enhanced disease resistance	[97]
	*M. oryzae*	Rice blast	*TMS5*, *Pi21*, and *Xa13*	Not defined	Yes	Enhanced disease resistance	[96]
	*M. grisea*	Rice blast	*OsMPK5*	Protoplast transformation	No	Resistance not confirmed	[98]
	Rice tungro bacilliform virus	Rice tungro disease	*eIF4G*	*Agrobacterium*-mediated transformation	Yes	Enhanced disease resistance	[99]
**Tobacco**	Cotton Leaf Curl Multan Virus (CLCuMuV)	DNA viral disease	IR and C1	*Agrobacterium*-mediated transformation	No	Complete resistance to virus infection	[100]
	Tomato Yellow Leaf Curly Virus (TYLCV), Beet Curly Top Virus (BCTV), and Merremia Mosaic Virus (MeMV)	DNA viral disease	IR, CP, RCRII	*Agrobacterium*-mediated transformation	No	No disease symptoms and reduced virus accumulation	[101]
	Bean Yellow Dwarf Virus (BeYDV)	DNA viral disease	LIR, Rep	*Agrobacterium*-mediated transformation	No	Reduced symptoms and virus load	[102]
	Beet Severe Curly Top Virus (BSCTV)	DNA viral disease	IR, CP, Rep	*Agrobacterium*-mediated transformation	No	Geminivirus-resistant plants	[85]
	Tomato Yellow Leaf Curl Virus (TYLCV)	DNA viral disease	CP, Rep	*Agrobacterium*-mediated transformation	No	Enhanced disease resistance	[103]
**Tomato**	*Pseudomonas syringae*	Bacterial speck	*SIDMR6–1*	*Agrobacterium*-mediated transformation	No	Enhanced disease resistance	[104]
	*P. capsici*	*Phytophthora* blight	*SIDMR6–1*	*Agrobacterium*-mediated transformation	No	Enhanced disease resistance	[104]
	*Xanthomonas* spp.	Bacterial spot	*SIDMR6–1*	*Agrobacterium*-mediated transformation	No	Enhanced disease resistance	[104]
	*P. syringae* pv. *tomato* *(Pto) DC3000*	Bacterial speck	*SlJAZ2*	*Agrobacterium*-mediated transformation	Not defined	Enhanced disease resistance and defence trade-off solved	[105]
	*O. neolycopersici*	Powdery mildew	*SlMlo1*	*Agrobacterium*-mediated transformation	No	Enhanced disease resistance	[64]
	*O. neolycopersici*	Powdery mildew	*SIPMR4*	*Agrobacterium*-mediated transformation	No	Enhanced disease resistance	[84]
	*Fusarium oxysporum* f. sp. *lycopersici*	Fusarium wilt	*Solyc08g075770*	*Agrobacterium*-mediated transformation	No	Enhanced disease susceptibility	[106]
	*B. cinerea*	Gray mold	*SlMAPK3*	*Agrobacterium*-mediated transformation	No	Enhanced disease susceptibility	[107]
	PVX, TMV	RNA viral disease	DCL2	*Agrobacterium*-mediated transformation	No	Resistance to PVX and TMV	[108]
	Tomato Yellow Leaf Curl Virus (TYLCV)	DNA viral disease	CP, Rep	*Agrobacterium*-mediated transformation	No	Enhanced disease resistance	[103]
**Wheat**	*Blumeria graminis* f. sp. *tritici*	Powdery mildew	*TaMlo*	Protoplast and Biolistic transformation	Yes	Enhanced disease resistance	[63]
	*B. graminis* f. sp. *tritici*	Powdery mildew	*TaEDR1*	Biolistic transformation	No	Enhanced disease resistance	[35]

**Table 3 biology-12-01037-t003:** Genes targeted using CRISPR-based genome editing techniques to impart tolerance of abiotic stress.

Abiotic Stresses	Plant Species	Targeted Gene	Delivery Method	Regulating Direction of Response to Stress Function	References
**Drought**	*Oryza sativa*	*OsNAC006*	*Agrobacterium* -mediated transformation	Transcription factor	[139]
	*Brassica napus*	*BnaA6.RGA*	*Agrobacterium*-mediated transformation	Transcription factor	[140]
	*O. sativa*	*SRL1*, *SRL2*	*Agrobacterium*-mediated transformation	Rolling of leaf	[141]
	*O. sativa* subsp. *indica*	*OsDST*	*Agrobacterium*-mediated transformation	Drought and salt tolerance (DST) gene	[119]
	*O. sativa*	*OsNAC14*	*Agrobacterium*-mediated transformation	Transcription factor	[142]
	*O. sativa*	*OsSAPK2*	*Agrobacterium*-mediated transformation	ABA signaling	[116]
	*O. sativa*	*OsMYB1*, *OsYSA*, *OsROC5*, *OsDERF1*, *OsPDS*, *OsPMS3*, *OsEPSPS*, *OsMSH1*, *OsMYB5*, *OsSPP*	*Agrobacterium*-mediated transformation	Amino acid synthesis	[143]
	*Solanum lycopersicum* L.	*SlNPR1*	*Agrobacterium*-mediated transformation	Drought tolerance	[117]
	*S. lycopersicum* L.	*MAPK3*	*Agrobacterium*-mediated transformation	Growth and development	[123]
	*Triticum aestivum*	*TaDREB2, TaERF3*	PEG-mediated transformation	Dehydration-responsive element-binding protein	[115]
	*Zea mays*	*ARGOS8*	Particle bombardment	Ethylene-responsive gene family regulator	[121]
	*Arabidopsis thaliana*	*AREB1*	*Agrobacterium*-mediated transformation	ABA signaling	[144]
	*A. thaliana*	*AtAVP1*, *AtPAP1*	*Agrobacterium*-mediated transformation	Transcription factor	[145]
	*A. thaliana*	*AtOST2*	*Agrobacterium*-mediated transformation	Stomatal movement	[146]
	*A. thaliana*	*AtMIR169a*, *AtMIR827a*, *TFL1*	*Agrobacterium*-mediated transformation	Negative factor of drought tolerance	[147]
	*Glycine max*	*GmMYB118*	*Agrobacterium*-mediated transformation	Transcription factor	[127]
	*Populus clone NE-19*	*PdNF-YB21*	*Agrobacterium*-mediated leaf disc method	Transcription factor ABA-mediated indoylacetic acid transport	[148]
**Cold**	*O. sativa*	*OsMYB30*, *OsPIN5b*, *GS3*	*Agrobacterium*-mediated transformation	Cold tolerance	[133]
	*A. thaliana*	*UGT79B2*, *UGT79B3*	*Agrobacterium*-mediated transformation	UDP-glycosyltransferases	[149]
	*O. sativa* subsp. *indica*	*OsPRP1*	*Agrobacterium*-mediated transformation	Plant growth and stress response	[150]
	*A. thaliana*	*AtCBF2*	*Agrobacterium*-mediated transformation	Encodes AP2/ERF (APETALA2/Ethylene-Responsive Factor)-type transcription factors)	[151]
	*A. thaliana*	*AtCBF1*, *AtCBF2*, *AtCBF3*	*Agrobacterium*-mediated transformation	Encodes AP2/ERF (APETALA2/Ethylene-Responsive Factor)-type transcription factors)	[147,152,153]
	*S. lycopersicum* L.	*SlCBF1*	*Agrobacterium*-mediated transformation	Transcription factor	[131]
	*O. sativa*	*OsAnn3*	*Agrobacterium*-mediated transformation	Plant development and protection from environmental stresses	[132]
	*A. thaliana*	*CBF*s	*Agrobacterium*-mediated transformation	Transcription factor	[154]
**Salinity**	*O. sativa*	*OsGTg-2*	*Agrobacterium*-mediated transformation	Transcription factor	[155]
	*O. sativa*	*PIL14*	*Agrobacterium*-mediated transformation	Phytochrome-Interacting Factor	[156]
	*O. sativa*	*OstPQT3*	*Agrobacterium*-mediated transformation	E3 ubiquitin ligase (enhances resistance to abiotic stresses)	[157]
	*O. sativa*	*OsAGO2*	*Agrobacterium*-mediated transformation	Transcriptional transactivator (growth and development, stress and defense responses, alternative splicing, and DNA repair)	[158]
	*O. sativa*	*OsDST*	*Agrobacterium*-mediated transformation	Zinc-finger transcription factor	[119]
	*O. sativa*	*FLN2*	*Agrobacterium*-mediated transformation	Sucrose metabolism fructokinase-like protein2	[159]
	*O. sativa*	*OsRR9* and *OsRR10*	*Agrobacterium*-mediated transformation	Cytokinin signaling	[160]
	*O. sativa*	*OsDOF15*	*Agrobacterium*-mediated transformation	Transcription factor (regulates cell proliferation in the root)	[161]
	*O. sativa*	*OsSPL10*	*Agrobacterium*-mediated transformation	Transcription factor	[162]
	*O. sativa*	*NCA1a*, *NCA1b*	*Agrobacterium*-mediated transformation	Regulation of catalase activity	[163]
	*O. sativa*	*RR22*	*Agrobacterium*-mediated transformation	Transcription factor (cytokinin signal transduction and metabolism)	[164]
	*O. sativa*	*OsNAC041*	*Agrobacterium*-mediated transformation	Transcription factor	[165]
	*O. sativa*	*OsOTS1*	*Agrobacterium*-mediated transformation	Salt stress response regulation	[166]
	*O. sativa*	*OsSAPK1*, *OsSAPK2*	*Agrobacterium*-mediated transformation	ABA pathway regulator	[129]
	*O. sativa*	*OsBBS1*	*Agrobacterium*-mediated transformation	Receptor-like cytoplasmic kinase	[167]
	*O. sativa*	*OsMIR408*, *OsMIR528*	*Agrobacterium* -mediated transformation	Salt stress response regulation	[168]
	*O. sativa*	*OsRAV2*	*Agrobacterium*-mediated transformation	Transcription factor	[126]
	*Z. mays*	*HKT1*	*Agrobacterium*-mediated transformation	High-affinity potassium transporter	[169]
	*A. thaliana*	*AtSAUR41*	*Agrobacterium*-mediated transformation	Auxin response gene	[170]
	*Cucurbita moschata*	*RBOHD*	*Agrobacterium*-mediated transformation	NADPH oxidase is a key member for H_2_O_2_ production	[171]
	*A. thaliana*	*AtC/VIF1*	*Agrobacterium*-mediated transformation	Cell wall/vacuolar inhibitor of fructosidases	[130]
	*Hordeum vulgare*	*HvITPK5/6*	*Agrobacterium*-mediated transformation	Sequential phosphorylation of inositol phosphate to inositol hexakisphosphate	[172]
	*G. max* L.	*GmAITR*	*Agrobacterium*-mediated transformation	Transcription factors that are involved in the regulation of ABA signaling	[173]
	*S. lycopersicum* L.	*SlARF4*	*Agrobacterium*-mediated transformation	ARFs play a key role in regulating the expression of auxin response genes	[129]
	*A. thaliana*	*ArathEULS3*	*Agrobacterium*-mediated transformation	Stress-responsive protein, stomatal closure	[174]
**Heat Stress**	*S. lycopersicum* L.	*BZR1*	*Agrobacterium*-mediated transformation	Brassinosteroid regulation	[136]
	*O sativa*	*OsPDS*	Gene gun	Phytoene Desaturase gene encodes one of the important enzymes in the carotenoid biosynthesis pathway	[175]
	*G. max* L.	*GmHsp90A2*	*Agrobacterium*-mediated transformation	Molecular chaperone and heat shock protein	[176]
	*O. sativa*	*OsHSA1*	*Agrobacterium*-mediated transformation	Chloroplast development at early stages and functions can protect chloroplasts under heat stress at later stages	[135]
	*S. lycopersicum* L.	*Slcpk28*	*Agrobacterium*-mediated transformation	Decreases the activity of antioxidant enzymes	[177]
	*O. sativa*	*OsNAC006*	PEG-mediated	Mediates the process of photosynthesis and limits the activity of antioxidant enzymes	[139]
	*Z. mays*	*TMS5*	Particle bombardment	Thermosensitive genic male sterile 5	[138]
	*Lactuca sativa*	*LsNCED4*	*Agrobacterium*-mediated transformation	Key regulatory enzyme in the biosynthesis of abscisic acid (ABA)	[178]
	*O. sativa*	*OsPYL1/4/6*	*Agrobacterium*-mediated transformation	ABA receptor	[179]

## Data Availability

No new data were created or analyzed in this study. Data sharing is not applicable to this article.

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
