# Peer review of "Recent Developments in CRISPR/Cas9 Genome-Editing Technology Related to Plant Disease Resistance and Abiotic Stress Tolerance"

_biology, 2023, doi:10.3390/biology12071037_

Round 1
Reviewer 1 Report
This manuscript mainly summarize the application of CRISPR/Cas9 technology in improving plant disease resistance and abiotic stress tolerance. Although the title is the recent developments on CRISPR/Cas9 genome editing technology for ... , this manuscript look more like a simple collection of CRISPR/Cas9 mediated cases for plant disease resistance and abiotic stress tolerance. The authors only compared CRIPSR/Cas system with TALENs and ZFNs, but did not mention any important improvement of the CRISPR/Cas genome editing tools, such as the optimization of PAM sequence requirement. Even so, this manuscript seemed to be written in early 2022 but not 2023, as all the papers cited in this manuscript are published before 2021.
Other comments:
1. The abstract didn't get to the points of the whole manuscript and should be rewritten.
2. The introduction took too long to come into subject.
3. In line 64-67, physical, chemical, and biological methods is not limited to gamma radiation, EMS, and T-DNA/transposon insertion.
4. The reference 36 related to NgAGO has been retracted and the statement and comment about this research should be prudent.
Minor editing of English language is required.
Author Response
This manuscript mainly summarize the application of CRISPR/Cas9 technology in improving plant disease resistance and abiotic stress tolerance. Although the title is the recent developments on CRISPR/Cas9 genome editing technology for this manuscript look more like a simple collection of CRISPR/Cas9 mediated cases for plant disease resistance and abiotic stress tolerance. The authors only compared CRIPSR/Cas system with TALENs and ZFNs, but did not mention any important improvement of the CRISPR/Cas genome editing tools, such as the optimization of PAM sequence requirement.
We have re-visited the section of the review and included information on base editing and prime editing with the relevant references.
Even so, this manuscript seemed to be written in early 2022 but not 2023, as all the papers cited in this manuscript are published before 2021.
We have updated several of the references to latest 2022/2023 papers.
Other comments:
1. The abstract didn't get to the points of the whole manuscript and should be rewritten.
This is now re-written.
2. The introduction took too long to come into subject.
We needed to cover the background to give the relevant information to the readers.
3. In line 64-67, physical, chemical, and biological methods is not limited to gamma radiation, EMS, and T-DNA/transposon insertion.
These are given as an example for those methods.
4. The reference 36 related to NgAGO has been retracted and the statement and comment about this research should be prudent.
We thank you for this, we did not know about the retraction, and we have read the retraction note and the NgAGO has been removed.
Comments on the Quality of English Language
Minor editing of English language is required.
The entire review has been attended.
Reviewer 2 Report
The authors delivered a comprehensive and well-organized review on the genome editing applied in plant biology. Most applications of genome editing are in animal for a potential therapeutic benefit in human. The authors addressed a gap in the field by providing a summary on the current successful approach in improving agricultural yield using genome editing. The summary table including specific strategies on disease and abiotic stress resistant provide a thorough reference for further study in the field. However, to improve the overall manuscript, it is better to include more graphic illustrations as mere table and text are sometimes dull to read.
Author Response
The authors delivered a comprehensive and well-organized review on the genome editing applied in plant biology. Most applications of genome editing are in animal for a potential therapeutic benefit in human. The authors addressed a gap in the field by providing a summary on the current successful approach in improving agricultural yield using genome editing. The summary table including specific strategies on disease and abiotic stress resistant provide a thorough reference for further study in the field. However, to improve the overall manuscript, it is better to include more graphic illustrations as mere table and text are sometimes dull to read.
We appreciate the reviewer’s supportive comments. We already have 2 Figures on the CRISPR and it would be overcrowded if we also put figures on the biotic and abiotic stresses.
Round 2
Reviewer 1 Report
The authors addressed most of the issues I concerned.